# The role of information communication technologies as a moderator of knowledge creation and knowledge sharing in improving the quality of healthcare services

**Simon Colnar[1], Ivan Radević[2]\*, Nikola Martinović[2], Anđelko Lojpur[2], Vlado Dimovski[1]**

**1** School of Economics and Business, University of Ljubljana, Ljubljana, Slovenia, **2** Faculty of Economics, University of Montenegro, Podgorica, Montenegro

\* radevic@ucg.ac.me

## Abstract

This study examines the role of knowledge creation, knowledge sharing and information communication technologies, which are organizational factors that influence the quality of healthcare services. In today's knowledge-intensive environment, understanding and gaining in-depth knowledge on how to improve the quality of healthcare services is gaining in importance and recognition. Quantitative data collected in 2019 with 151 respondents employed in healthcare organizations was used. Running a series of hierarchical linear regression models, we found a significant positive relationship between knowledge creation and quality of healthcare services, and a significant positive relationship between knowledge sharing and quality of healthcare services. Empirical data additionally provides support for information communication technologies that act as a moderator both in the relationship between knowledge creation and knowledge sharing with quality of healthcare services. With our data, we provide empirical backing for the impact of knowledge creation, knowledge sharing and information communication technologies on the quality of healthcare services that are provided by Montenegrin healthcare organizations. Our paper offers theoretical and practical implications derived from our research study.

## 1. Introduction

Information and communication technologies are identified as one of the crucial enablers of knowledge management practices and most relevant and contemporary literature suggests that appropriate technology solutions within organizations are of significant importance in relation to successful knowledge management initiatives [1]. In today's knowledge intensive world of work [2], the concept of knowledge management is becoming increasingly important as a tool that may be vital to a higher level of organizational effectiveness. Ongoing growing importance of information communication technologies has already changed traditional forms of organizational functioning, which consequently determined the concept of knowledge management to become integral as a tool for achieving higher levels of organizational

**Data Availability Statement:** All relevant data are within the paper and its Supporting Information files.

**Funding:** This work was supported by the Slovenian Research Agency, Program P5-0364 – The Impact of Corporate Governance, Organizational Learning, and Knowledge Management on Modern Organization. The funders had no role in study design, data collection and analysis, decision to publish, or preparation of the manuscript.

**Competing interests:** The authors have declared that no competing interests exist.

effectiveness [3]. This change in terms of the approach paradigm, emphasizes even more the role of knowledge as a determining factor of improved organizational performance [4]. In essence, this concept implies a process of efficient and effective learning, through research, exploitation and sharing of human knowledge, with a support of adequate technological advancements [5]. Previous research posits the concept of knowledge management as a determinant of organizational success [6], improved service quality [7] and as a tool that enables organizations to make internal improvements [8, 9].

Nowadays, the field of healthcare services is continuously exposed to pressures from different stakeholders to improve the quality of its services [10]. Moreover, researchers Parand et al. [11] suggest that a number of challenges related to quality of healthcare services remain unsolved and require the attention of both academics and practitioners. Therefore, it becomes crucial to gain in-depth knowledge and understanding regarding healthcare service quality dimensions and define actions that could help healthcare services providers with improving their overall organizational effectiveness [12]. There is a wide range of industries, where the application of knowledge management can result in positive improvements of organizational performance, including healthcare services, where the knowledge of employees represents the core of providing care for patients. Moreover, an adequate knowledge management process results in the adoption of quality decisions by healthcare professionals, and in better outcomes for patients [13]. In addition, the adequate knowledge management process in healthcare organizations is also vital for raising the level of healthcare services in practice [14]. With our paper, we aim to respond to calls of researchers to enhance the knowledge regarding the concept of quality of healthcare services [15] and to gain additional understanding of knowledge management as applied in the healthcare environment [16].

In current state-of-the-art research there is a gap in considering the impact of specific organizational factors such as knowledge creation, knowledge sharing and information communication technologies on the quality of healthcare services. Additional insight is of paramount importance as it enables healthcare services providers to gain knowledge about potential activities and solutions for improving the quality of healthcare services [11]. As existing theory seems to be difficult to apply within the healthcare environment, we focus within the scope of our paper on improving the understanding and gaining further knowledge of the construct of knowledge creation, where researchers Boon Sin et al. [17] claim that knowledge creation leads to improved organizational performance, which applies also to public sector organizations, including healthcare institutions. In a similar vein, we intend to strengthen previous research by providing further insight into the impact of knowledge sharing on achieving higher levels of organizational performance [18], which is in this paper explored as the quality of healthcare services. Previous research validates the positive relationship between information communication technologies and enhanced organizational performance [19], where we aim to provide additional insight into information communication technologies and their impact on improving the quality of healthcare services.

The purpose of our paper is to add to contemporary research, by theoretically proposing a conceptual model and empirically testing the impact of organizational factors that influence knowledge management activities within the healthcare sector as part of the ongoing attempts to enhance the quality of healthcare services [20]. We investigate the relationship between knowledge creation and quality of healthcare services and knowledge sharing and quality of healthcare services. With our research, we are able to add to previous theoretical findings within the context of the knowledge management discipline that posit that individual knowledge management activities are typically in a positive relationship with organizational performance [21]. In a similar vein, we add to contemporary knowledge management theory that recognizes knowledge as a critical resource for the functioning of organizations [22], including

healthcare organizations as their performance is being inextricably tied to efficient use of knowledge [23]. In addition, we explored for the potential moderating effect of information communication technologies on the aforementioned proposed relationships. We utilized a quantitative analysis of collected data from healthcare employees to test our hypotheses in healthcare institutions in Montenegro. As we obtained data for all our variables in a one-time single survey, we acknowledge that common method bias might be a methodological issue in our study. With our paper, we aim to contribute towards advancing the body of literature of quality of healthcare services and knowledge management in the context of healthcare organizations [24], where currently a gap in knowledge exists [15, 25]. Moreover, our research offers theoretical guidance to healthcare employees that rely on information-communication technologies to cope with a continuously increasing need to manage knowledge [26] and extends the research on success determinants of knowledge management within the healthcare setting. As such, our research is one of the few studies that explore the effects of organizational factors on knowledge management initiatives in healthcare organizations as suggested by Ali et al. [27]. Similarly, our study is following the proposition of Siong et al. [28] that the role of knowledge and knowledge management is attracting increased attention from scholars and practitioners as an effort to achieve organizational excellence.

Following our overarching theory of the knowledge-based view of the organization [29–31], we emphasize the important role of knowledge in healthcare organizations as we propose that knowledge management is one of the primary sources that influence the functioning of such organizations and subsequently has the potential to increase the quality of healthcare services that are offered to users in practice. Previous research within the knowledge-based view of the organization [29, 32] suggests that the presence and right utilization of knowledge has the potential to lead to higher levels of organizational performance [33]. Similarly, contemporary research of Martin and Javalgi [34] posits that the attainment and deployment of knowledge is integral for organizational performance. In addition, we aim to empirically clarify the relationship between organizational factors that impact knowledge management and its relationship with organizational performance [35].

In line with McIver and Lepisto [21], our study provides further insight into the knowledge management discipline aspect that is focused on findings that enable organizations to achieve competitive advantage in practice by managing and utilizing what they know or even what they will need to know. Such a state is particularly useful in the contemporary knowledge intensive economy. Another learning outcome for practitioners might be that knowledge-based organizations should not only blindly apply knowledge management related initiatives as they need to align them with activities on how to motivate, support and offer relevant knowledge to individual employees for knowledge initiatives to be successful. In such a situation, the role of managers is integral as they need to motivate their employees to be engaged and utilize available resources to improve their individual and organizational performance [36]. As healthcare can be considered a practice-based profession, the goal of knowledge management in practice would be aimed to add value to services and to increase social wellbeing, societal effectiveness and general welfare [37, 38].

## 2. Literature review

Healthcare systems, as well as micro-level health facilities, generally depend on data and information collected by patients, medical doctors, or obtained from scientific studies [39, 40]. In this context, management of information, knowledge creation and knowledge sharing, are key areas in the healthcare system. An improved knowledge management system contributes to better decisions of healthcare professionals and results in better treatment outcomes as a result

of the healthcare provided [27]. The extent to which knowledge creation and knowledge sharing contribute to better treatment outcomes is one of the key information that we seek to obtain.

Information-communication technology systems with their support to knowledge management processes positively influence the competitiveness of organizations [1]. As internet technology enables rapid search, access, exchange and retrieval of information it is deemed as suitable for collaboration and knowledge exchange between organizational members [41]. Moreover, such systems typically support knowledge management practices as they facilitate knowledge acquisition and creation, knowledge dissemination, knowledge conversion and knowledge utilization [42, 43]. Typically, technology is viewed an essential component and integral facilitator in any knowledge management initiative [44]. With appropriate training and education for employees, such solutions have become crucial to organizations as they carry out many tasks related to knowledge management [45].

Previous research of Lopez-Nicolas and Soto-Acosta [46] supports a significant positive impact of having an appropriate information-communication technology infrastructure on knowledge creation. In a similar vein, Sambamurthy and Subramani [47] have highlighted the critical role of information-communication technologies in shaping organizational efforts for knowledge creation. In addition, researchers Davenport and Prusak [48] and Roberts [49] posit that information-communication technology is a crucial aspect of knowledge creation due to the fact that such technologies facilitate speedy collection, storage and exchange of knowledge. While knowledge creation can be conducted without the support of information-communication technologies, such technology allows knowledge creation to be performed in a more effective manner [50]. As state-of-the-art research supports the fact that information-communication technologies are an important enabler of the knowledge creation process, such technology must be designed an utilized in a manner that it is aligned with other organizational resources, with a particular emphasis on human resources, namely employees [51].

In previous research, information-communication technology was identified as an organizational factor that influences knowledge sharing at individual and team level [52]. In contemporary research, information-communication technology has been proposed as one of the crucial enablers of knowledge sharing [53]. Information-communication technology can also have a potentially significant influence on the knowledge sharing activity [54] as it provides the infrastructure that enables the establishment, maintenance, and intensification of relationships within and among teams [55]. Contemporary information-communication technology systems that include also social networks can help employees to share their knowledge through common platforms and enable electronic storage of information and knowledge. Furthermore, information-communication technology systems can facilitate collaboration between employees and teams, and enrich their communication through various modern tools [44]. Moreover, information-communication technology can accelerate access to information and knowledge that is stored in databases to enhance the knowledge sharing process in organizations. Such technology can support knowledge sharing by enabling effective communication channels and tools and by identifying the source of information or knowledge [52]. In a similar vein, Riege [56] is of the opinion that information-communication technology enables instant access to large amounts of information and knowledge to facilitate long-distance collaboration and knowledge sharing between employees and organizations.

## 2.1. Knowledge creation and quality of healthcare services

Knowledge creation is a continuous process that implies ongoing interaction between individuals and groups at the organizational level [57–59]. Moreover, knowledge creation as a process

consists of four stages: socialization, externalization, combination, and internalisation [60]. Creation of new knowledge can similarly be the basic source of competitive advantage over longer periods of time [61] and is useful for any organization, whether public or private. Creating a comprehensive system that enhances the process of creating new knowledge, helps the organization achieve its strategic goals [30]. Knowledge creation is an important aspect of development and implementation of high-quality services and products [62]. As such it is perhaps even the most crucial aspect in the complex environment of healthcare, where evidence typically gained from empirical research supports the efficiency of utilizing limited available resources [63]. Moreover, knowledge creation is of the primary activities of knowledge intensive organizations [64], including healthcare organizations. Healthcare organizations heavily rely on clinical knowledge for delivering services in practice [65]. Therefore, healthcare employees who create and sharing clinical knowledge have a paramount role in knowledge management activities in healthcare organizations [66, 67]. In addition, appropriately implementing knowledge creation in practice is especially complicated in the healthcare environment that is often characterized by evidence-based practice, where making decisions on how to provide or improve healthcare is related to integrating best available research evidence with a combination of clinical expertise and patient values, knowledge and preferences [68]. In the area of healthcare services, the strategic goal is higher quality of services provided, through a combination of efforts of healthcare professionals and direct interaction with patients [69]. Consensus on health service quality indicators has not yet been reached in the literature [70], and it is recommended that each institutions should develop their own system of indicators. However, the quality of health care is increasingly observed through reports on the performance of health systems in different countries, ie. through a system of organizational performance [71, 72]. Definitions of quality of health care common to all stakeholders imply effective care that contributes to patient satisfaction [73]. Knowledge creation and its inter-organizational dissemination, through the use of an adequate network concept [74] and the necessary data cataloguing, contributes to positive repercussions on the organizational performance of healthcare providers. Since knowledge creation is a continuous process, its constant improvement is vital for the benefit of all stakeholders. Knowledge creation can have a significant positive impact on professional development of employees in the environment of healthcare institutions [75]. This study extends previous research that argues that the creation of knowledge is a crucial aspect of providing quality services in healthcare practice [76]. In a similar vein, our study extends existing research on knowledge creation as the final outcome of the process that enhances the quality and quantity of healthcare organization's knowledge base [77], which subsequently influences the quality of services that are provided in practice to users. In addition, modern development of technology is a significant accelerator of the process of knowledge creation at the organizational level [78, 79]. Consequently, this combination of the healthcare system and information communication technologies has changed the way healthcare is provided and it contributed to greater benefits for patients [80]. In spite of the existence of solid scientific research, aimed at analysing the relationship between the concept of knowledge creation and organizational performance [81], with the influence of information communication technologies as a newly associated scalar value [82], the subject area has not been analysed adequately within the concept of quality of healthcare services in the healthcare system of Montenegro. Moreover, our research answers the calls of research to provide suggestions on how can healthcare organizations can effectively deal with complex challenges such as knowledge creation for the successful and quality functioning of the healthcare system as a whole [77]. This paper, based on empirical research, aims to close this gap. In view of this, our first hypothesis is:

*Hypothesis 1: Knowledge creation is positively related to the quality of healthcare services*

## 2.2. Knowledge sharing and quality of healthcare services

Knowledge sharing in organizations is defined as the process through which individuals, groups, departments or the whole organization are affected by the experience and knowledge of another [83]. Knowledge sharing within healthcare organizations is recognized as one of the main indicators of quality, innovation, competitiveness, growth and development of the organization. Healthcare can be considered as a patient-centered environment, where healthcare professionals have to continuously cooperate with experts from several fields such as nurses, social workers and many others. In such a state, the contemporary and relevant knowledge has to be effectively management and shared among healthcare employees to improve the quality of services. Therefore, efficient knowledge sharing activities are crucial for healthcare organizations [84]. Authors Shahmoradi et al. [14] highlight that knowledge sharing plays a particularly important role in knowledge management activities within the evidence-based practice that is common within the healthcare setting. As healthcare organizations have a direct impact on people's quality of life and wellbeing, the effectiveness of knowledge sharing is even more important in organizations that function in such a delicate environment [85]. Crass and Peters [86] further highlight the complexity of knowledge in healthcare as they posit that the majority of innovations and the delivery of services are reliant on the skills and know-how of employees in healthcare organizations. The importance of the impact of knowledge sharing on the quality of healthcare services has caused the recent growth of interest in the area [87]. Existing studies similarly identified quality information exchange as one of the key indicators of quality of services within healthcare organizations in addition to the competences of healthcare professionals [88–90]. Moreover, cross-national studies have indicated the importance of organizational culture as a component that influences the willingness of health professionals to be active participants in the process of knowledge sharing [91]. A system that is too centralized negatively affects the process of knowledge sharing at the organizational level [92], so it is necessary to find the right balance also in this aspect. The existence of an optimal level of theoretical knowledge and practical skills and competences of healthcare professionals are key indicators of successful knowledge sharing among healthcare professionals [91]. Improving the quality of healthcare services is ever more based on the improvement of both knowledge creation and knowledge sharing [93]. In addition, globally there is a growing social, political and social interest in the exchange of knowledge and experience in the context of improving healthcare, while emphasizing the key role of scientific community and medical staff in the process of generating new value and new knowledge [94]. With our study, we add to previous research that argues that healthcare teams frequently consist of interdisciplinary members that need to transfer knowledge to one another to be able to increase existing knowledge and create new knowledge and adequate healthcare solutions, which in practice improves the quality of services that are provided [95]. Historically, knowledge transfer in the healthcare environment was hindered due to numerous reasons. In the future, academics and practitioners are aiming to find improvements in knowledge sharing activities among healthcare professional as it can represent a tool to ultimately improve the quality of healthcare services [96]. Therefore, based on previous research, our paper further deepens the analysis of the subject area through the study of knowledge sharing and the quality of healthcare services within the Montenegrin healthcare system. The second hypothesis arises from the above:

*Hypothesis 2: Knowledge sharing is positively related to the quality of healthcare services*

## 2.3. Information communication technologies and quality of healthcare services

Information communication technologies, through the use of computers, the Internet, mobile devices and various interactive platforms, significantly shape the functioning of modern organizations, their systems, processes and communication [97]. Advanced information communication technologies help the process of knowledge creation through a number of different functionalities such as analysis and presentation, data storage and management, networking and communication, as well as interaction and collaboration [79, 98–100]. Creating new knowledge and innovation has become crucial in the process of implementing information communication technologies in the regular practice of healthcare institutions [101]. The creation of information and creation of new knowledge is the area where information communication technologies, may contribute to a higher quality of healthcare services [102]. Nowadays, the ability to effectively access needed information and to distinguish between relevant and irrelevant information is becoming an ever important skill for professionals and organizations [103]. In this context, Soto-Acosta and Cegarra-Navarro [45] emphasize the role of information communication technologies in the exploitation and management of existing knowledge. The ability of individuals, professionals and organizations to have access to and later to disseminate health related information in today's electronic society emphasizes the necessity of adopting information communication technologies [104] within the healthcare environment. Additionally, this possibility for creating new knowledge, through information communication technologies, enables the application of the concept "*patient-centered care*", which implies directly and positively a higher quality of services provided [105]. Andreeva and Kianto [1] also point to the specific role of information communication solutions at the level of organizational knowledge, organizational performance, and organizational competitiveness. Our study adds to existing state-of-the-art literature that provides insight into the fast growing field of information-communication technologies that support the successful functioning of the healthcare sector with the provision of safe, efficient, high-quality, and information-communication technology assisted healthcare services. The provision of such services is typically reliant on an adequate workforce, financial resources and the knowledge and know-how on how to utilize services [106]. Moreover, our research builds the body of literature that promotes the positive impact of information-communication technology adoption on the safety, efficiency and quality of services in healthcare organizations [107] within the specific context of the Montenegro healthcare environment. In this paper, we tested the concept of improving the quality of healthcare services, through the moderating effect of information communication technologies, defining the third hypothesis as follows:

***Hypothesis 3***: *Information communication technologies moderate the positive relationship between knowledge creation and quality of healthcare services*

In addition to knowledge creation, knowledge sharing is an area where information communication technologies contribute to the sharing of good practice in healthcare. Information communication technology platforms, in various forms and shapes, enable the knowledge sharing among healthcare professionals [85]. The appropriate utilization of information communication technologies contributes to faster and better inter-organizational knowledge sharing, all to the benefit of patients as end users [108]. The application of various tools and applications enables a quicker transfer of knowledge both between service providers and between providers and users of healthcare services, thus raising awareness of the importance of healthcare [109]. This can play a very important role also in providing remote healthcare, through the use of technology and rapid exchange of information related to the medical state

of patients, implementation of adequate diagnostics, treatments and disease remediation [110]. With our paper, we build on the notion that technological factors as for example information-communication infrastructure, technological skills, knowledge management tools and availability of technological facilities partially define the success of such interventions within the healthcare environment. Moreover, such technological factors are becoming more and more important in the successful provision of high-quality healthcare services in practice [111]. Moreover, with our research we aim to close the gap of understanding on how information-communication technology can be used to promote the quality of services provided for patients [112]. In a similar vein, we add to the body of literature that emphasizes the connection between information-communication technology and quality of healthcare services, which was a field that was partially neglected in the past due to an emphasis on technological interventions that offer mainly financial benefits [113]. We tested empirically the understanding of the moderating effect of information communication technologies in the relationship between knowledge transfer and quality of healthcare services, by setting the following hypothesis:

> *Hypothesis 4: Information communication technologies moderate the positive relationship between knowledge sharing and quality of healthcare services*

We present our conceptual model with the above mentioned hypotheses in Fig 1.

## 3. Methodological approach

### 3.1. Data collection procedure and measurement

The process of collecting primary data involved the use of a questionnaire to understand the process of knowledge creation, knowledge sharing and information communication technologies within healthcare organizations with a special focus on the quality of services provided. The complete process of data collection was done in accordance with ethical rules, norms and strict scientific research protocols valid at the University of Montenegro. To begin the process, we requested the Committee for Ethical Issues at the University of Montenegro's consent and from competent bodies at the Faculty of Economics and the Ministry of Health of Montenegro. The Committee (for Ethical Issues) at the University of Montenegro declared itself

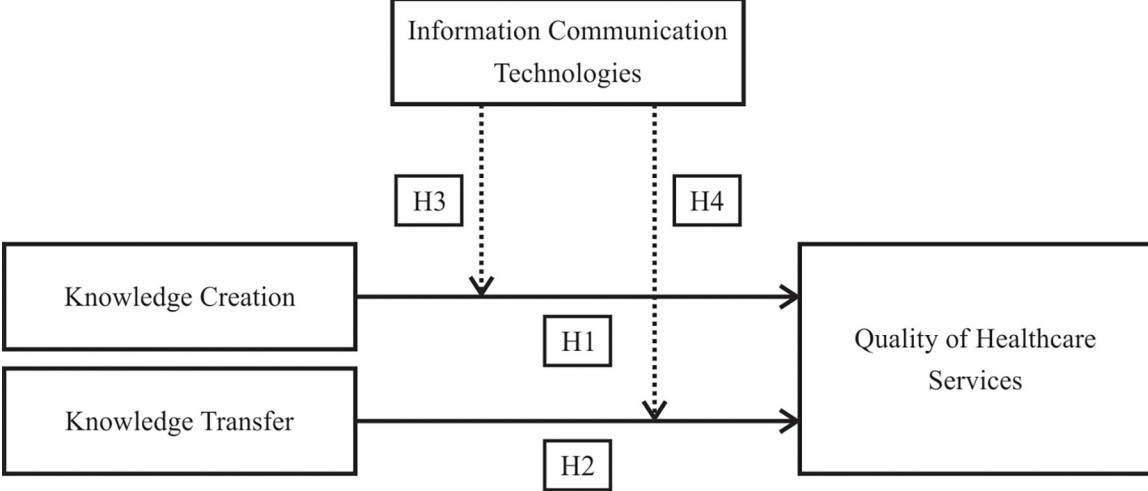

**Fig 1. Conceptual model of the relationships between knowledge creation, knowledge sharing, quality of healthcare services and information communication technologies.**

incompetent to give such consent, unlike the Faculty of Economics and the Ministry of Health, which provided their consents in writing. Having completed that step, respondents were contacted to take part in the survey, via e-mail and/or telephone. After the respondents consented to participate in the survey, we agreed on the interview date and time i.e. during the work-hours. In our research, participation was entirely voluntary and anonymous. Immediately before the interviews began, the interviewer informed the respondents in detail about the purpose of the study, applied scientific and research protocols, and asked for their explicit consent. Once the interviewers signed the consent in writing, the interviewer began with the data collection from the respondents (note: all respondents are adults, working in the health sector). All consents given in writing and duly signed by respondents were archived in the Faculty Archives Unit following the successful completion of the research under applicable Faculty regulations. Our survey(s) did not include any opportunities to identify the individual responses and link it to the respondents' identities. A sample of 45 health care institutions in Montenegro was generated by a combination of institutions from public (32) and private sector (13). Subsequently, the sample represents health institutions of primary, secondary and tertiary levels of health care. Besides, the sample included health care institutions from all three regions of the country (center, north and south). Specifically, it includes 18 institutions from the central part, 14 from the northern part and 13 from the southern part of Montenegro.

In the process of selecting an appropriate sample, researchers face certain dilemmas. Namely, random sampling is traditionally considered as the gold standard in order to achieve sample impartiality, which is directly against the background of obtaining reliable estimates [114]. Yet the trade-off between the desire to randomize the sample on the one hand, and pragmatization on the other, is one of the leading challenges in the decision making process of the subject issue. Constraints on time, resources as well as rising costs, imply that random sample selection is not always a realistic option. Conventional or purposive sampling [115] is emerging in the field of health care research as one of the solutions to this problem. The justification of this approach is confirmed by existing studies that support the representativeness of the sample defined on the basis of such strategies [116].

The data were collected in May 2019, and 151 healthcare workers took part in the survey. In order to ensure the principles of impartiality and non-selectivity and the concept of comprehensiveness, information was collected across various organizational levels in healthcare institutions. Accordingly, the sample consists of 45 directors, 45 medical doctors, 45 technicians, and 16 members of the Union of Medical Doctors, which altogether makes a total of 151 individuals whose opinion was taken into account. Just over three-fifths of the sample are women, and the remaining two-fifths are men. When it comes to age structure, respondents aged between 50 and 59 are a dominant portion of the sample (37.5%), followed by respondents aged 30 and 39 (24.3%), and respondents aged 40 and 49 (20.1%).

Majority of healthcare workers, 94.7%, have worked in the healthcare system for more than five years, 93.3% have worked for more than five years for their current employer (i.e. medical organization), hence the sample is representative in terms of the respondents' ability to realistically perceive the processes of knowledge creation and knowledge sharing, information communication technologies and quality of healthcare services in the context of analysed variables of this study.

## 3.2. Methods

To analyze whether our results might be affected by common method biased, we applied Harman's one-factor test [117]. The obtained results indicate that the first factor makes 56.7% of the total variance. This result is slightly above the recommended value threshold (50%) by

Podsakoff et al. [118], suggesting that common method bias might be a limiting factor in the study. The obtained data were imported within the SPSS 25.0 version.

In order to analyse each individual research construct, the authors used measuring instruments that have a high frequency of use and are adequately validated in contemporary scientific research. All measuring instruments meet a predefined set of criteria: they are often cited in research papers published in relevant scientific journals, they are up-to-date in the sense that they are used in the most recent research, and finally, they are well conceptually established in the context of their frequent use by key authors from our research scope [7, 119–124].

**3.2.1. Knowledge creation.** Two item scale was used to determine this construct (α = .90), which is adapted by Downes [120]. The measurement of this variable was conducted by measuring the degree of agreement of the respondents with the following items: "*My organization has mechanisms for creating or acquiring knowledge from different sources such as volunteers, clients, donors or competitors*".

**3.2.2. Knowledge sharing.** This construct was observed through the use of eight item scale (a = .93) adapted by Downes [120], that he used to measure knowledge sharing. The questionnaire involves answering items such as "*In my organization, it is easy to identify key experts in certain areas and learn how to get in touch with them*".

**3.2.3. Quality of healthcare services.** As with the knowledge sharing analysis, the Downes [120] scale that consists of 3 items (α = .87) was used. Downes [120] adapted it to measure the quality of healthcare services. The questionnaire consists of the following items: "*Within my organization, we provide higher quality services to our customers*" and "*All in all, our organization works better*".

**3.2.4. Information communication technologies.** A two item scale was used in this case (α =. 67) that was adapted by Downes [120]. It is based on statements examining whether technical support to employees is always available, as well as whether employees are confident enough to use information communication technologies or avoid using them due to lack of experience. The questionnaire comprises of items such as: "*Technical support for information systems is readily available*".

**3.2.5. Control variables.** As for control variables, there are two control variables that make an integral part of our research: age and the highest level of education. In research, the decision to include or exclude control variables may have implications for drawing final conclusions based on the research conducted [125]. Against the background of the individual and existing knowledge management reserach, demographic characteristics, such as age and the highest level of education, may have an impact on the overall level of knowledge management activities in an organization, which is the subject of analysis [126]. It is important to note that both control variables that are considered control in our research have already been the subject of analyses in researches that covered knowledge management [127].

To explore the convergent validity of all items utilized to measure constructs in our research we examined standardized factor loadings [128]. In Table 1, we report the range of our standardized factor loadings in our measurement model. Standardized factor loadings for all of our four constructs were statistically significant (> .50). One item intended to measure

**Table 1. Confirmatory factor analysis results.**

| Construct | No. of items | Reliability (Cronbach alpha) | Range of standardized coefficients (factor loadings) | CRI | AVE |
|---|---|---|---|---|---|
| Knowledge Creation | 2 | 0.90 | 0.82 to 0.86 | 0.83 | 0.71 |
| Knowledge Sharing | 8 | 0.93 | 0.65 to 0.82 | 0.90 | 0.54 |
| Quality of Healthcare Services | 3 | 0.87 | 0.64 to 0.92 | 0.83 | 0.62 |
| Information Communication Technologies | 2 | 0.67 | 0.72 to 0.84 | 0.76 | 0.61 |

knowledge creation and two items intended to measure quality of healthcare services and information communication technologies did not meet the criteria recommended in the literature and were therefore omitted from the final model. Our final model consists of 15 items utilized to evaluate the existing state of four measured constructs. To test the composite (constructs) reliability we explored the composite reliability index (hereinafter: CRI) and average variance extracted (hereinafter: AVE) [129]. To fulfill research criteria, we follow the suggested values of Diamantopolous and Sigaw [130], which are for AVE (.40) and CRI (.60). We present AVE and CRI values for our measured constructs in Table 1. Numerous fit indices to evaluate the model fit to data at the global level exist [131].

## 4. Results

Fit indices are as follows: CFI = 0.97; chi-square: 104.690; RMSEA = .07; and df = 67 and are satisfactory (without modification indices, the results of the model fit were: CFI = .84, chi-square = 296.722, RMSEA = .14, and df = 84.). Selected descriptive statistics for our measured variables are presented in Table 2. Respondents on average value quality of healthcare services (4.06) the best in their organization, followed by knowledge sharing (3.90) and knowledge creation (3.71). The construct of information communication technologies received a significantly lower evaluation (2.40). Between our measured variables the correlation coefficients are moderately or strongly positive with ranges between .65 and .88 and moderately or weakly negative with ranges between -.18 and -.28. A significant and positive correlation was evident between knowledge sharing and quality of healthcare services (.71; p < 0.01) and knowledge sharing and knowledge creation (.88; p < 0.01). In addition, there was a significant and negative correlation between knowledge sharing and information communication technologies (-.28; p < 0.05). Quality of healthcare services displayed a significant and positive correlation with knowledge creation (.65; p < 0.01) and a significant and negative correlation with information communication technologies. In the scope of our research, knowledge creation had a significant and negative correlation with information communication technologies (-.25; p < 0.01). Between our two control variables there is no significant correlation.

Hypothesis 1 (H1) explored the direct relationship between knowledge creation and quality of healthcare services. Hypothesis 2 (H2) examined the direct relationship between knowledge sharing and quality of healthcare services. In hypothesis 3 (H3), we include information communication technologies as a moderator of the relationship between knowledge creation and quality of healthcare services. Similarly, in hypothesis 4 (H4) we include information communication technologies as a moderator of the relationship between knowledge sharing and quality of healthcare services. We ran a series of hierarchical regression analysis utilizing centered

**Table 2. Mean values, standard deviations and coefficient correlations (n = 151).**

| Variable | Mean | SD | 1 | 2 | 3 | 4 | 5 |
|---|---|---|---|---|---|---|---|
| 1. Age | 48.82 | 10.76 | | | | | |
| 2. Highest Level of Education | 4.72 | 0.64 | -0.06 | | | | |
| 3. Knowledge Sharing | 3.90 | 0.94 | 0.11 | 0.01 | | | |
| 4. Quality of Healthcare Services | 4.06 | 0.86 | -0.01 | -0.05 | 0.71** | | |
| 5. Knowledge Creation | 3.71 | 1.10 | 0.05 | -0.01 | 0.88** | 0.65** | |
| 6. Information Communication Technologies | 2.40 | 0.85 | -0.01 | -0.12 | -0.28** | -0.18* | -0.25** |

**p < 0.01 and

*p < 0.05

SD = Standard Deviation, p = significance

**Table 3. Hierarchical regression analysis predicting quality of healthcare services–models 1–4.**

| Variables | Model 1 | | | | Model 2 | | | | Model 3 | | | | Model 4 | | | |
|---|---|---|---|---|---|---|---|---|---|---|---|---|---|---|---|---|
| | b | s.e. | ß | t | b | s.e. | ß | t | b | s.e. | ß | t | b | s.e. | ß | t |
| Age | -.004 | .005 | -.055 | -.766 | -.002 | .005 | -.028 | -.412 | -.004 | .005 | -.048 | -.685 | -.002 | .005 | -.029 | -445 |
| Highest Level of Education | -.166 | .091 | -.132 | -1.817 | -.161 | .086 | -.128 | -1.869 | -.165 | .089 | -.131 | -1.848 | -.177 | .083 | -.141 | -2.131* |
| C_KNOC | .471 | .057 | .604 | 8.258** | | | | | .464 | .056 | .595 | 8.344** | | | | |
| C_KNOT | | | | | .560 | .059 | .658 | 9.538** | | | | | .542 | .057 | .637 | 9.561** |
| C_ICT | -.071 | .127 | -.041 | -.556 | -.092 | .121 | -.053 | -.764 | .084 | .137 | .049 | .616 | .083 | .127 | .048 | .650 |
| C_KNOCxC_ICT | | | | | | | | | .242 | .090 | .208 | 2.699** | | | | |
| C_KNOTxC_ICT | | | | | | | | | | | | | .304 | .091 | .240 | 3.339** |
| $R^2$ | 0.369 | | | | 0.436 | | | | 0.404 | | | | 0.483 | | | |
| F(df) | 17.96(123) | | | | 23.76(123) | | | | 6.61(122) | | | | 22.81(122) | | | |
| $\Delta R^2$ | 0.369 | | | | 0.436 | | | | 0.035 | | | | 0.047 | | | |

*$p < 0.05$

**$p < 0.01$

variables to test our conceptual model that is explained with our proposed hypotheses in Fig 1. In our first direct effect model (model 1), we include knowledge creation as the independent variable and age and highest level of education as our control variables. Within model 2, we include knowledge sharing as the independent variable and the aforementioned control variables. In addition, in our third model (model 3) we explore the suggested two-way interaction effect between knowledge creation and information communication technologies. In model 4, we include our second proposed interaction effect between knowledge sharing and information communication technologies. We present a more in-depth analysis of our four models in Table 3.

We found a significant and positive relationship in model 1, between knowledge creation (β = .60; exact p = .000) and quality of healthcare services. We are able to provide empirical support for H1 with our data. In addition, we also found a significant and positive relationship between knowledge sharing (β = .66; exact p = .000) and quality of healthcare services in model 2. Therefore, we are also able to provide empirical support for H2 on the basis of our data. In models 3 and 4 we included information communication technologies as the moderator of knowledge creation (model 3) and knowledge sharing (model 4) with quality of healthcare services. Both models showed considerable added value in relation to the direct effect models as expressed in model 1 and 2. The $R^2$ change is .035 in model 3 in comparison with model 1 and .047 in model 4 in comparison with model 2.

Our results in model 3 show a significant and positive relationship on the example of our two-way interaction effect between knowledge creation and information communication technologies and on the quality of healthcare services (β = .21; exact p = .008). In line with the above, we are able to provide empirical support also for our H3. Similarly, model 4 indicates a significant and positive relationship of our two-way interaction effect consisting of knowledge sharing and information communication technologies with quality of healthcare services (β = .24; exact p = .001). With our results, we are to support H4 on the basis of empirical data. Additionally, we present the simple slope analysis of both H3 and H4. The simple slope analysis for H3 indicates it is significant (exact p = .000). Moreover, we present the interaction between knowledge creation and information communication technologies and their influence on quality of healthcare service (see Fig 2).

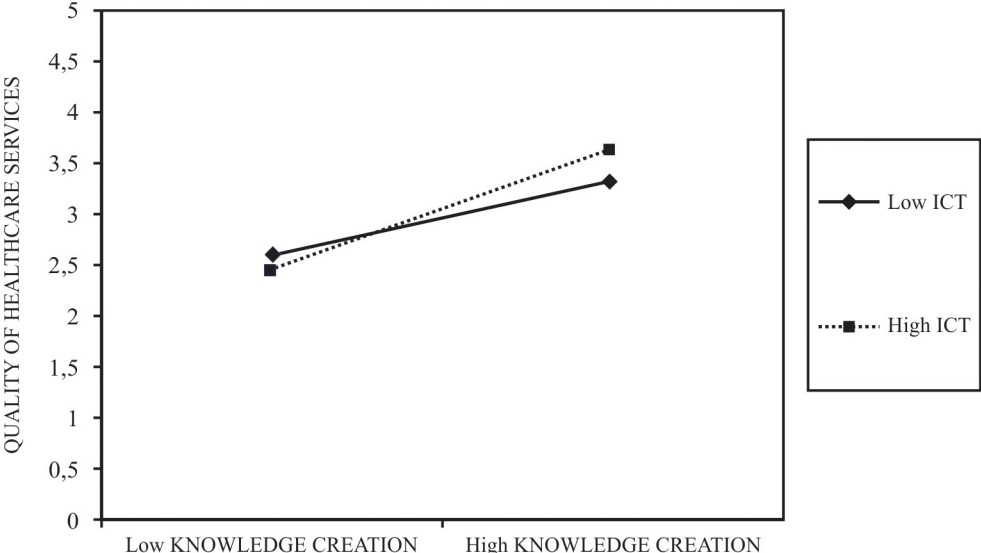

**Fig 2. Interaction between knowledge creation and information communication technologies in influencing quality of healthcare services.**

Highest levels of quality of healthcare services occur when the level of information communication technologies is high. In addition, the impact of knowledge creation is similarly important as both in the case of low information communication technologies and high information communication technologies, higher levels of knowledge creation indicate to better quality of healthcare services. In the example of high knowledge creation, best quality of healthcare services is also related to high information communication technologies, while the comparison to low information communication technologies shows a significant difference. In the example of low knowledge creation, low information communication technologies imply better quality of healthcare services.

Moreover, the analysis of the simple slope for H4 is also significant (exact p = .000). The graphical representation of the interaction effect between knowledge sharing and information communication technologies as they influence the quality of healthcare services is presented in Fig 3.

The highest levels of quality of healthcare services are related to high levels of information communication technologies. Moreover, also the influence of knowledge sharing is important as higher levels of knowledge sharing positively influence the quality of healthcare services. When levels of knowledge sharing are low, low information communication technologies contribute to better quality of healthcare services. Interestingly, when levels of knowledge sharing are low, the combination with higher levels of information communication technologies produces worse results in terms of quality of healthcare services.

## 5. Discussion

This paper involved structuring a conceptual research model that is composed of four constructs: knowledge creation, knowledge sharing, information communication technologies and the level of quality of provided healthcare services. The study implied setting up of four hypotheses in order to analyse the relationship between the defined variables, through the implementation of a hierarchical linear regression model. The first hypothesis starts from the assumption that knowledge creation is in a direct positive correlation with the level of quality

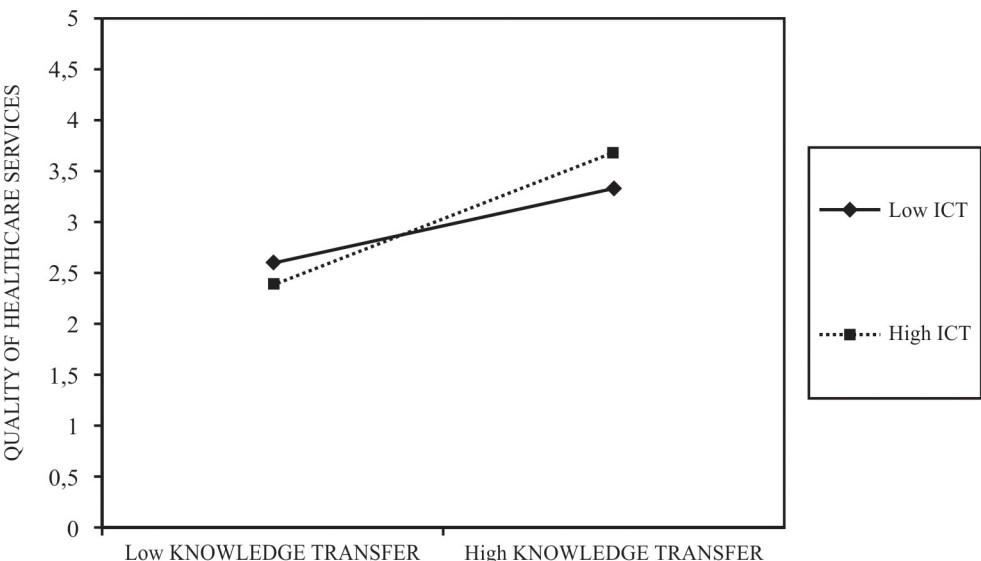

**Fig 3. Interaction between knowledge sharing and information communication technologies in influencing quality of healthcare services.**

of healthcare services. The second hypothesis also implies a positive correlation between knowledge sharing and the quality of healthcare services, while the third and fourth examine the moderating role that information communication technologies play in fostering knowledge creation and knowledge sharing that lead to better performance of healthcare providers.

The results of the conducted research confirm the existence of a direct positive correlation between knowledge creation and the quality of healthcare services (H1). In other words, knowledge creation is vital in determining the quality of healthcare provided to patients. Such results are in accordance with previous research, which argued that knowledge creation leads to improved organizational performance [17]. Similarly, our results further validate the findings of Ayanbode and Nwagwu [75] and Kitson et al. [74], that knowledge creation has a positive impact on the organizational performance also on the example of healthcare organizations.

The results obtained on the basis of the study confirmed the second hypothesis, the existence of a positive correlation between knowledge sharing and the level of quality of healthcare services provided (H2). Knowledge sharing enables the dissemination of best medical practices and a better output in terms of healthcare provided to patients. Even in situations where it is applicable, the valuable knowledge gained from medical research is essentially useful only if used by all stakeholders within the healthcare system. Unfortunately, healthcare providers in practice often lack information about current trends, latest scientific knowledge and researches that applied best medical practice, and are not aware how their application could be a deciding factor in treatment of patients. Under such circumstances, patients are hindered as they do not receive the best possible treatment, or receive the one that does not match entirely their specific health condition [132]. Therefore, the sharing of newly created knowledge and medical practice is an important determinant of healthcare provided. Our results support the claim of Lombardi [18] that knowledge sharing influences the achievement of higher levels of organizational performance. In a similar vein, our results are in line with the suggestions of Wensing and Grol [93] that knowledge creation has an important impact on improving the quality of healthcare services.

The results of our conducted research provided empirical support to the third hypothesis. Information communication technologies moderate the positive correlation between knowledge creation and the level of quality of healthcare services (H3). With our results we are able to further validate the opinion of Tripathi et al. [102] that information communication technologies act as a moderator in the relationship between knowledge creation and higher quality of healthcare services. Similarly, we add to state-of-the-art research of Papanastasiou et al. [105] and Andreeva and Kianto [1] that emphasize the specific role of information communication technologies on the improvement of healthcare services.

Finally, this study confirmed that information communication technologies also play an important role in facilitating the sharing of knowledge, and that they contribute to a higher level of quality of healthcare services (H4). The higher level of knowledge sharing is closely related to the quality of healthcare services, and information communication technologies are the factor that is vital to the progress in this field. Such findings correlate to existing literature that stresses the impact of information communication technologies on knowledge sharing [85] and its subsequent influence on improving the quality of healthcare services [108].

Information communication technologies enable the knowledge sharing and the dissemination of good practices at various different organizational levels. Universities, teaching hospitals and research institutes remain the main hubs of sources for the creation and sharing of newly created knowledge in the field of healthcare. The use of information communication technologies and documenting best practices in healthcare institutions requires constant work on improving and updating the whole system. This can be a challenge, given the speed of technological change and the growing need for rapid dissemination of new knowledge, especially in times of crisis, such as the current global COVID-19 pandemic. In that context, COVID-19 introduced an exponential threat to the theory and practice of quality [133]. If the research was hypothetically conducted today, it is very possible that despite ICT positive implications, we would find somewhat different results related to the process of creating and disseminating knowledge. Bearing in mind that the fight against the pandemic is still an ongoing battle, that information is changing on a daily basis, the quality of the knowledge created can be very questionable. Researches are done in a very fast way, by urgent procedure, and even the created know-how, despite the fact that it is encouraged by the use of the most sophisticated ICT equipment, can be short-lived. When it comes down to knowledge sharing, major pharmaceutical companies, guided primarily by lucrative goals, are willing to make a drug or vaccine against the virus available at an adequate price, but are not willing to make available the formula or know-how used in the process of creating the necessary medicament. In this way, research conducted during the pandemic era would probably conclude that the application of ICT technologies has contributed to the creation of knowledge (the issue of quality may be questionable), but not so much to its dissemination. In any case, on the example of COVID-19 pandemic we can assume the following in relation to our research constructs: first, creating knowledge in a certain way is a condition of survival (finding a vaccine), knowledge sharing is important not only at the institutional level (micro aspect) countries (cooperation between health institutions and knowledge sharing), but also between countries in order to control the global problem such as a pandemic; thirdly, ICT is an important element of organizational design (noticeable growing importance of telemedicine) and fourthly, health, ie the quality of health services is imposed as a priority on which other areas of human social activity depend, and is expected to be increasingly in the focus of theorists and practitioners in the field of management and organization. The organizational goal must be to create a system that will allow medical doctors and other healthcare professionals to know at all times where to find up-to-date information related to a specific field of medicine, or to access newly available knowledge [134].

There are several theoretical contributions of our research. First, with our results we contribute to existing research that focuses on the knowledge-based view of the organization [29–31] as we emphasize the integral role of knowledge within the environment of healthcare organizations. Furthermore, we add to up-to-date research by theoretically proposing a conceptual model and empirically testing organizational factors of knowledge management that have the potential to increase the quality of healthcare services [20]. By testing and validating our model on primary data, we are able to contribute to a broader understanding and further insight on knowledge and knowledge management [29, 32] and the positive influence on organizational performance [33, 34]. Second, with our research we respond to some of the existing challenges that are related to the area of quality of healthcare services [11] as we provide empirical insight into actions that could help services providers with improving quality of their healthcare services as was suggested in Tripathi and Siddiqui [12]. Third, we add to the debate of knowledge creation and its influence on the advancement of organizational performance [17, 74, 75]. With our research, we extend the understanding of the aforementioned relationship on the example of healthcare institutions. Fourth, we provide additional support to suggestions that knowledge sharing has a positive impact on improving the overall organizational performance [18] in our case understood as the quality of healthcare services. With this part of our research, we validate the opinion of Wensing and Grol [93] that improving the quality of healthcare services is also related to the levels of knowledge sharing within an organization. Fifth, with our results we are able to promote the positive impact of information communication technologies on improved organizational performance as was emphasized in the research of Yunis et al. [19]. In addition, our results are in line with Tripathi et al. [102], where the authors suggest that information communication technologies as a moderator can contribute towards achieving higher levels of quality of healthcare services in practice. Sixth, with our study we are able to extend the understanding of knowledge management in the context of the public sector [24, 25], where we focused our research in exploring knowledge management within the healthcare environment, which was typically overlooked in previous studies.

The practical implications of our research are intended for managers, practitioners and decision makers and are identified as opportunities for improvements within the healthcare system in Montenegro, through better understanding and knowledge of organizational factors such as knowledge creation, knowledge sharing and information communication technologies. Those organizational factors are instruments that can positively influence the levels of quality of healthcare services. The content and suggestions that can be derived from our research in the form of concrete recommendations can help stakeholders engaged in the healthcare system to create appropriate conditions for achieving better organizational performance, which will subsequently raise the quality of healthcare services. Hopefully, on the basis of our results, managers and employees in healthcare organizations will devote more attention, resources and efforts towards implementing activities and initiatives that include knowledge creation, knowledge sharing and information communication technologies with the final aim of improving the overall quality of healthcare services. Relying on empirical evidence, our study offers the opportunity or starting point for interested stakeholders to improve their knowledge, skills and competences related to knowledge management and providing quality healthcare services in practice. Nevertheless, it is important that managers are aware of the relevant alignment of their knowledge management activities within their specific organizational context and also in relation to their available knowledge management resources. In addition, within the specific and complex healthcare environment, identified and assessing appropriate knowledge management activities might be even more challenging.

In spite of the numerous theoretical and practical contributions of our paper, some limitations exist. First, the methodological issue of common method bias as revealed by Harman's

single factor test [117] is present. Second, as our results are based on a sample from only one country, we argue that it would be beneficial to conduct a cross-national study to provide a higher degree of generalization of our findings. Third, we have to take into account the complexity of the healthcare environment, which might negatively influence the responses from our respondents as they are constantly exposed to demanding and draining situations at work. Fourth, within this research we did not distinguish between some of the factors determining the characteristics of healthcare organizations such as the size of the organization that could be measured with the number of its employees. Fifth, in general in the healthcare environment, measuring the quality of healthcare services is difficult as we are predominantly relying on the perception of either healthcare services providers or healthcare services users.

Given the limitations of our study, we identified additional opportunities for future research on the topic of quality of healthcare services, including the following: (1) to counteract the potential negative effect of common method bias, we propose to collect the data for the dependent, independent and moderating variables at different points in time; (2) we would advise to focus on conducting similar research on an international sample; (3) to mitigate the negative effect of potential employee bias, we should include a higher number of respondents in future studies; (4) future research should include additional control variables that are measured at the organizational level; (5) it is necessary to promote international efforts to standardize the measurement of quality of healthcare services, which would additionally enable the comparison between different countries; (6) considering the complexity of the health care system, it would be useful to apply a qualitative research approach in addition to the quantitative one, ie. to use a mixed-method approach, in order to obtain an in-depth explanation, with full use of triangulation methods.

## 6. Conclusion

The proposed conceptual model and the conducted empirical study within our research is aimed at examining the impact of knowledge creation and knowledge sharing on the level of quality of provided healthcare services. Thus, our study contributes towards the expansion of the conceptual framework of knowledge management within the healthcare environment and aims to support ongoing efforts to improve the overall quality of healthcare services in practice. More specifically, we explored the moderating effect that information communication technologies might have on the existence of the two predefined relationships. Obtained results clearly indicate a direct and positive link between knowledge creation and quality of healthcare services and between knowledge sharing and quality of healthcare services. Moreover, information communication technologies further moderate the relationships that are important in influencing the quality of healthcare services. Highest levels of quality of healthcare services occur when the level of information communication technologies is high. In a nutshell, the impact of knowledge creation and knowledge sharing is important as higher levels of knowledge creation and knowledge sharing lead to better quality of healthcare services. Additionally, our study proposes some promising directions to conduct future research on this topic.

## Supporting information

**S1 Data.**
(XLSX)

## Author Contributions

**Conceptualization:** Simon Colnar, Ivan Radević, Nikola Martinović.

**Data curation:** Ivan Radević.

**Formal analysis:** Simon Colnar, Nikola Martinović.

**Funding acquisition:** Vlado Dimovski.

**Investigation:** Ivan Radević.

**Methodology:** Simon Colnar, Nikola Martinović.

**Project administration:** Ivan Radević, Vlado Dimovski.

**Resources:** Ivan Radević.

**Software:** Nikola Martinović.

**Supervision:** Anđelko Lojpur, Vlado Dimovski.

**Validation:** Simon Colnar, Nikola Martinović.

**Writing – original draft:** Simon Colnar, Nikola Martinović.

**Writing – review & editing:** Simon Colnar, Ivan Radević, Anđelko Lojpur, Vlado Dimovski.

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
