## [Decision Letter · Decision Letter 0]

4 Nov 2021

PONE-D-21-25036The role of information communication technologies as a moderator of knowledge creation and knowledge transfer in improving the quality of healthcare servicesPLOS ONE

Dear Dr. Radević,

Thank you for submitting your manuscript to PLOS ONE. After careful consideration, we feel that it has merit but does not fully meet PLOS ONE’s publication criteria as it currently stands. Therefore, we invite you to submit a revised version of the manuscript that addresses the points raised during the review process.

We look forward to receiving your revised manuscript.

Kind regards,

Nadja Damij, Ph.D.

Academic Editor

PLOS ONE

Reviewers' comments:

Reviewer's Responses to Questions

**Comments to the Author**

1. Is the manuscript technically sound, and do the data support the conclusions?

Reviewer #1: Yes

Reviewer #2: Partly

2. Has the statistical analysis been performed appropriately and rigorously? 

Reviewer #1: Yes

Reviewer #2: I Don't Know

3. Have the authors made all data underlying the findings in their manuscript fully available?

Reviewer #1: Yes

Reviewer #2: No

4. Is the manuscript presented in an intelligible fashion and written in standard English?

Reviewer #1: Yes

Reviewer #2: Yes

5. Review Comments to the Author

Reviewer #1: Dear Author(s),

Thank you for submitting your paper for PLOS ONE. I believe your paper has good merit and potential to be published once you revise the paper.

1) A general comment about relevancy. We are all faced with the same issue, that after the pandemic nothing is the same any more, could you actualise on that?

2) Some other research (https://blogs.bmj.com/bmj/2019/05/01/cheng-hock-toh-current-geographical-spread-research-failing-patients/) have hypothesised, that there is a big discrepancy between centre and periphery. Might this be an explanatory factor also in your research?

You will find my comments in the attached PDF.

Kind regards.

Reviewer #2: 1. The paper addresses an interesting and relevant topic. Overall, the paper is well-organised with good readability.

2. However, the contributions are unclear in terms of originality and implications to theory and practice within the knowledge management discipline. The title of the paper suggests an enabling role of ICT for knowledge management. Yet, a rich discussion on the role of ICT infrastructure (platforms, tools, etcetera), digital skills and governance in fostering knowledge creation and sharing is missing.

3. The literature review does not reveal the gaps in existing research that justify the paper's originality.

4. The complexity of the knowledge creation/sharing issues in healthcare services, which are typically led by evidence-based knowledge and practices by clinicians/experts, is missing in the literature review.

5. The quality of healthcare services and organisational performance are often used interchangeably. The literature review does not sufficiently address the quality (or performance) aspects in a healthcare setting.

6. Line 119: In describing the four stages of knowledge creation (SECI model), the word 'internationalisation' needs to be replaced by 'internalisation'.

7. Since the healthcare services primarily deal with expertise-driven tacit knowledge, 'knowledge sharing' is probably a more appropriate term to use instead of knowledge transfer.

8. The paper refers to ICT as an organisational factor on several occasions (line 15, 65, 456). Do the authors mean the ICT maturity or capability as an organisational factor to influence the knowledge management activities?

9. For the research constructs, the authors mentioned using well-established measuring instruments frequently employed in contemporary research. However, the absence of a few key references indicates methodological weakness.

10. The study involves survey participants from a combination of institutions across public and private sectors (public: 32 and private: 13). It would have been interesting to see whether an organisation's characteristics (public vs private) affect its knowledge management activities.

6. PLOS authors have the option to publish the peer review history of their article (what does this mean?). If published, this will include your full peer review and any attached files.

Reviewer #1: No

Reviewer #2: No

---

## [Author Response · Author response to Decision Letter 0]

2 Feb 2022

Answer to the editor: Dear editor, we would like to thank you and the reviewers for your valuable feedbacks which helped us to improve our manuscript. We believe that the progress made in the manuscript is also heavily influenced by numerous valuable suggestions provided from our reviewers in the revision process. We hope you will find our responses satisfactory.

Answer to the reviewer 1: We would like to thank you for your precise revision. We have improved part of the text according to your valuable suggestions. Your review enabled us to solve many important issues that needed to be addressed within the scope of our paper. Your clear instructions were of paramount importance to improve our paper, with our intention to fully follow your recommendations. Thank you for your effort and precise revision.

Answer to the reviewer 2: We would like to thank you for your precise and insightful revision. We have rewritten part of the text according to your valuable propositions. You helped us a lot by opening many important topics that needed to be addressed in our research. Your instructions were integral to partially restructure the whole paper, with our intention to fully follow your comments. Sincerely thank you for your effort and clear instructions.

---

## [Decision Letter · Decision Letter 1]

19 Jul 2022

PONE-D-21-25036R1

The Role of Information Communication Technologies as a Moderator of Knowledge Creation and Knowledge Sharing in Improving the Quality of Healthcare Services

PLOS ONE

Dear Dr. Radević,

Thank you for submitting your manuscript to PLOS ONE. After careful consideration, we feel that it has merit but does not fully meet PLOS ONE’s publication criteria as it currently stands. Therefore, we invite you to submit a revised version of the manuscript that addresses the points raised during the review process.

We look forward to receiving your revised manuscript.

Kind regards,

Nadja Damij, Ph.D.

Academic Editor

PLOS ONE
---

## [Editor Report · Acceptance letter]

26 Jul 2022

PONE-D-21-25036R1 

The Role of Information Communication Technologies as a Moderator of Knowledge Creation and Knowledge Sharing in Improving the Quality of Healthcare Services 

Dear Dr. Radević:

I'm pleased to inform you that your manuscript has been deemed suitable for publication in PLOS ONE. Congratulations! Your manuscript is now with our production department. 

Kind regards, 

on behalf of

Dr. Dragan Pamucar 

Academic Editor

PLOS ONE